# Catalysis of Organic Pollutants Abatement Based on Pt-Decorated Ag@Cu_2_O Heterostructures

**DOI:** 10.3390/molecules24152721

**Published:** 2019-07-26

**Authors:** Xiaolong Zhang, Bingbing Han, Yaxin Wang, Yang Liu, Lei Chen, Yongjun Zhang

**Affiliations:** 1Key Laboratory of Functional Materials Physics and Chemistry of the Ministry of Education, Jilin Normal University, Changchun 130103, China; 2National Demonstration Center for Experimental Physics Education, Jilin Normal University, Siping 136000, China; 3College of Chemistry, Jilin Normal University, Siping 136000, China

**Keywords:** catalysis, Ag@Cu_2_O–Pt, organic pollutants

## Abstract

Pt-decorated Ag@Cu_2_O heterostructures were successfully synthesized using a simple and convenient method. The Pt nanoparticle density on the Ag@Cu_2_O can be controlled by changing the concentration of the Pt precursor. The synthesized Ag@Cu_2_O–Pt nanoparticles exhibited excellent catalytic performance, which was greatly affected by changes in the Ag@Cu_2_O–Pt structure. To optimize the material’s properties, the synthesized Ag@Cu_2_O–Pt nanoparticles were used to catalyze toxic pollutants and methyl orange (MO), and nontoxic products were obtained by catalytic reduction. The Pt-decorated Ag@Cu_2_O nanoparticles showed excellent catalytic activity, which significantly decreased the pollutant concentration when the nanoparticles were used for catalytic reduction. The redistribution of charge transfer is the nanoparticles’ main contribution to the catalytic degradation of an organic pollutant. This Pt-decorated Ag@Cu_2_O material has unique optical and structural characteristics that make it suitable for photocatalysis, local surface plasmon resonance, and peroxide catalysis.

## 1. Introduction

The use of nano-semiconductor particles as a catalyst has a theoretical basis that, on the one hand, the quantum size effect broadens the semiconductor energy gap, the conduction band potential becomes more negative, and the valence band potential becomes more positive, which gives the catalyst a stronger redox capacity. On the other hand, the specific surface area of the nanoparticles is much larger than that of conventional materials; a nanometer-sized nanomaterial has a surface area equivalent to that of a football field. Nanomaterials have a good ability to adsorb pollutants, which is very advantageous for increasing the speed of the catalytic reaction. Moreover, the smaller the particle size, the smaller the probability of recombination of electrons and holes and the better the charge separation effect, resulting in the improvement of catalytic activity [1,2].

Noble metals and metal oxides are often used as materials in gas sensors, chemical coatings, biological detection, and catalysis because of their unique properties [3,4]. Because of the limitations of the available single-metal and metal-oxide materials, their properties are relatively simple and cannot meet the needs of actual applications. Thus, many researchers have undertaken the synthesis and characterization of new materials. In these studies, it was found that combinations of noble metals and metal oxides show many excellent properties and have better properties than single metals or metal oxides in microelectronics, biological detection, catalysis, and other applications [5,6,7]. In addition, metal and metal-oxide composite structures have more advantages than single components. Moreover, the catalytic performance of metal–metal oxide nanoparticles (NPs) can be adjusted by controlling the composition and architecture of the particles [8,9]. A number of metal–semiconductor composites have been synthesized, such as Au@Cu_2_O, Ag@Cu_2_O, ZnO–Pt, Pt–TiO_2_, and Cu_2_O@Pt [10,11,12,13,14]. Metal–semiconductor nanocomposites exhibit excellent catalytic properties. Using a metallic material as a core and modifying the surface with a metal oxide are a commonly used and feasible strategy for the production of conventional catalyst carrier materials [15]. Compared with metals or semiconductors, metal–semiconductor composites have many new properties, including a plasmon Synergistic effect, plasmon-induced catalysis, and resonance energy transfer, that are easy to control [16,17].

The noble metal in a heterogeneous precious metal catalyst is highly dispersed in the form of particles on the support, which can be a metal oxide or a molecular sieve or the like. This allows for a catalyst with better performance, such as Au@Cu_2_O, Ag@Cu_2_O, ZnO–Pt, Ag/TiO_2_, Pt–TiO_2_, and Cu_2_O@Pt, that combines the different properties of the two materials. The semiconductor itself has high catalytic activity, but it also has limitations in many reactions. A catalyst that is prepared by depositing a metal with a certain catalytic activity on the surface of the semiconductor can effectively separate photogenerated electrons and holes, reduce the overvoltage of the reduction reaction, and greatly improve the activity of the catalyst [10,11,12,13,14]. In the process of the formation of metal–metal and metal–semiconductor materials, the charge is usually rearranged. In the process of charge transfer, the properties of the materials will also significantly change, which is convenient for studying the properties of the materials [18]. For catalysis, charge transfer is the main catalytic mechanism. Methyl orange is often used as an adsorbent and catalytic target to characterize the catalytic performance of materials because it is a common indicator in the laboratory and easy to detect [19]. Additionally, 4-nitrophenol (4-NP) is often used as an intermediate of specialized chemicals, such as pesticides, medicines, and dyes. Because 4-NP is a common pollutant, several catalytic materials have been developed to treat it [20]. An Fe_3_O_4_–Pd nanocomposite was designed for use in catalytic applications and to be recycled [21]. An Ag–Au heterostructure was used to identify the catalytic mechanism of 4-NP [22]. Cu_2_O–Ag was prepared by controlling the reaction between Cu_2_O and AgNO_3_ in a simple way; the Cu_2_O–Ag showed a good ability in transforming organic pollutants in water [23]. Au@meso-SiO_2_ nanoparticles were synthesized as catalysts and exhibited excellent catalytic performance in the reduction of 4-p-nitrophenol in the presence of NaBH_4_ [24].

Cu_2_O has excellent optical and electrical properties that have attracted the attention of scientists [25]. Trinitrophenol (TNP), an explosive pollutant, is distributed widely in soil and water. The catalytic degradation of TNP in soil and water is one of the problems that needs to be addressed in environmental protection [26]. Herein, we synthesized Ag@Cu_2_O–Pt nanoparticles for photocatalysis. Ag@Cu_2_O nanoparticles were synthesized by reducing Cu(NO_3_)_2_ with hydrazine hydrate using polyvinylpyrrolidone (PVP) as the polymerization agent. PtCl_6_^−^ was reduced to Pt nanoparticles that were decorated on Ag@Cu_2_O because its reduction can lower the surface energy of the system. We controlled the density of the Pt nanoparticle coating by changing the Pt precursor concentration, and this structural change was used to adjust the catalytic performance of Ag@Cu_2_O–Pt. The larger the number of contact points between Ag@Cu_2_O and Pt, the more free electrons are available and the higher the probability that the catalytic target will be randomly adsorbed onto Ag@Cu_2_O–Pt. More importantly, the excellent adsorption ability of Ag@Cu_2_O–Pt was found to significantly enhance its catalytic activity toward 4-NP and methyl orange (MO).

## 2. Materials and Methods

### 2.1. Materials

Silver nitrate (AgNO_3_), Chloroplatinic acid (H_2_PtCl_6_∙6H_2_O), copper nitrate trihydrate (Cu(NO_3_)_2_∙3H_2_O), polyvinylpyrrolidone (PVP), sodium citrate (Na_3_C_6_H_5_O_7_∙2H_2_O), methyl orange (MO), 4-p-nitrophenol (4-NP), hydrazine hydrate (H_4_N_2_∙H_2_O), sodium borohydride (NaBH_4_), and anhydrous alcohol were purchased from Shanghai Sinopharm Chemical Co., Ltd., Ultra-pure water (18 MΩ·cm) was used throughout the experiment.

### 2.2. Preparation of Ag@Cu_2_O–Pt Nanocomposites

Ag nanoparticles. Silver nanoparticle sols were synthesized by the reduction of AgNO_3_ in the presence of Na_3_C_6_H_5_O_7_∙2H_2_O. A 200 mL 0.01% AgNO_3_ solution was placed in a three-necked flask and magnetically stirred (600 rpm) until it was slightly boiling. Then, 4 mL of a 1% Na_3_C_6_H_5_O_7_∙2H_2_O solution was added dropwise. The color gradually turned from light yellow to grey green. Afterwards, the solution continued to be heated for 40 min; finally, the silver sol was obtained. The spherical Ag nanoparticles were approximately 35–45 nm in size.

Ag@Cu_2_O nanoparticles. Ag@Cu_2_O was prepared according to a previously reported method [27]. PVP (1 g) was added to 50 mL of a 0.01 M Cu(NO_3_)_2_ aqueous solution and then magnetically stirred (300 rpm). Afterwards, 16 mL of the Ag nanoparticle solution was added, followed by 34 μL of H_4_N_2_∙H_2_O (35 wt%). The color of the solution changed within a few seconds of the addition of Ag nanoparticles, forming the Ag@Cu_2_O core-shell nanoparticles. The final color of the solution depended on the amount of Ag nanoparticles that was added. The color was changed quickly, and the product was washed with absolute ethanol and then dispersed in absolute ethanol and stored at 4 °C. The thickness of the Cu_2_O shell is adjustable and depends on the added amount of Ag nanoparticle solution.

Ag@Cu_2_O–Pt nanoparticles. To prepare the Ag@Cu_2_O–Pt nanoparticles, the dried Ag@Cu_2_O nanoparticles were dispersed in ultra-pure water (50 mL). Then, 5 mL of different concentrations of H_2_PtCl_6_ were added to 5 mL of the Ag@Cu_2_O nanoparticles to form the final 10 mL solution. The concentrations of H_2_PtCl_6_∙6H_2_O were 0.95 × 10^−4^, 1.26 × 10^−4^, 1.43 × 10^−4^, 1.52 × 10^−4^, 1.58 × 10^−4^, and 1.63 × 10^−4^ M. The solutions were mixed and stirred for 2 min. Once the color of the solution had changed from brown to light yellow, the Pt nanoparticles were successfully coated onto the Ag@Cu_2_O nanoparticles. The products were washed with absolute ethanol and ultra-pure water three times and then dried for 6 h at 60 °C.

### 2.3. Catalytic Reduction

In a cuvette, 2 mL of Ag@Cu_2_O–Pt nanoparticles and 0.1 mL of 0.005 mol/L 4-NP were added, followed by the quick addition of a freshly prepared NaBH_4_ (0.2 mol/L) solution. The solutions were colorless and bright yellow. Subsequently, 5 μL of either a Ag@Cu_2_O solution or a Ag@Cu_2_O–Pt solution (prepared by adding 0.95 × 10^−4^, 1.26 × 10^−4^, 1.43 × 10^−4^, 1.52 × 10^−4^, 1.58 × 10^−4^, and 1.63 × 10^−4^ M H_2_PtCl_6_∙6H_2_O) was mixed to study the catalytic reduction.

### 2.4. Characterization

The chemical compositions of the samples were characterized by X-ray photoelectron spectroscopy (XPS, Thermo Scientific ESCALAB 250Xi A1440 system, Thermo Fisher Scientific, Waltham, MA, USA). Transmission electron microscopy (TEM) images were obtained using a Hitachi H-800 (JEOL 2100, JEOL Ltd., Tokyo, Japan) transmission electron microscopy at an acceleration voltage of 200 kV. The UV-Vis absorption spectra and catalytic properties were monitored using a SHIMADZU 3600 spectrometer (Shimadzu Corporation, Tokyo, Japan).

## 3. Discussion

### 3.1. Properties of the Ag@Cu_2_O–Pt Heterostructure Nanocomposites

Cu_2_O nanocrystals were coated onto the Ag core and grew into Ag@Cu_2_O nanoparticles. The thickness of the Cu_2_O shell is negatively correlated with the volume of Ag sol. In the present study, the average size of the Ag@Cu_2_O particles is 105 nm (Figure 1a). Each Ag@Cu_2_O nanoparticle contained either a single silver core or a few silver cores or was a coreless particle. Pt nanoparticles were prepared on the surface of Ag@Cu_2_O by in situ reduction [28]. The in situ reduction of Ag@Cu_2_O with chloroplatinic acid (H_2_PtCl_6_∙6H_2_O) at room temperature did not involve the use of surfactants or capping agents. Because the size of the grown Pt nanoparticles was very small, the growth of Pt on the Ag@Cu_2_O surface cannot be directly observed. Presented in Figure 1b–g are the Ag@Cu_2_O surfaces coated with different densities of Pt nanoparticles. We observed that there was no significant change. Figure 2 shows the Ag@Cu_2_O–Pt elemental scan image of Ag@Cu_2_O with the addition of 1.58 × 10^-4^ M H_2_PtCl_6_∙6H_2_O. It can be seen that Pt is coated on the Ag@Cu_2_O surface by elemental scanning of the image plane.

Figure 3 shows optical pictures of the Ag@Cu_2_O and different Ag@Cu_2_O–Pt solutions. As shown in Figure 3, the color of the solutions changed from brown to yellow with increasing Pt precursor concentration. In Figure 3, the absorbance spectra of the Ag@Cu_2_O and different Ag@Cu_2_O–Pt solutions show complex optical characteristics. For the nanoshells, the spectral region above 500 nm was induced by the local surface plasmon resonance (LSPR) of the materials [29]. As the concentration of Pt increases, the local dielectric constant increases, which leads to a slight redshift of the absorption peak. The color change of the Ag@Cu_2_O–Pt solutions in Figure 3 further supports the redshift of the surface plasmon resonance (SPR) [7]. When the amount of the Pt precursor that was added to the Ag@Cu_2_O solution was between 0.95 × 10^−4^ and 1.63 × 10^−4^ M, the density of the Pt coating on Ag@Cu_2_O increased, and a color change from light yellow to light green occurred with increasing density of the Pt shell.

The surface components of the nanoparticles were detected by XPS. Figure 4 shows the binding energy spectra of the XPS measurements, corrected to the carbon peak (C 1s = 284.04 eV). The XPS spectrum obtained from Ag@Cu_2_O–Pt (the Pt precursor concentration was 1.58 × 10^-4^ M) shows that the main peaks are those of Pt 4f and O 1s, as shown in Figure 4A. The binding energies of the Pt 4f and O 1s peaks are 73.88 eV and 531.84 eV, respectively, for Ag@Cu_2_O–Pt (the Pt precursor concentration was 1.58 × 10^−4^ M). Figure 4B(a) shows the Cu 2p XPS spectral region for Ag@Cu_2_O [30]. Figure 4B(b–g) shows the Cu 2p XPS spectral region of the Ag@Cu_2_O–Pt samples with different amounts of Pt. The binding energy of the Cu 2p_3/2_ peak in the Ag@Cu_2_O–Pt (the Pt precursor concentrations were 0.95 × 10^−4^, 1.26 × 10^−4^, 1.43 × 10^−4^, 1.52 × 10^−4^, 1.58 × 10^−4^, and 1.63 × 10^−4^ M) samples shifted slightly relative to that of the Cu_2_O nanoparticles, indicating that the charge distribution in the Ag@Cu_2_O–Pt sample had changed and a complex charge transfer had occurred [28]. These results show that an interface had formed between Ag@Cu_2_O and Pt.

### 3.2. Catalytic Activity of the Ag@Cu_2_O–Pt Heterostructure

The Pt-coated Ag@Cu_2_O nanoparticles were prepared by in situ reduction. Compared with the Ag@Cu_2_O structure, the Pt-coated Ag@Cu_2_O nanoparticles had a more porous structure. The strong absorption of the Ag@Cu_2_O–Pt nanoparticles makes this material attractive for use in reducing toxic substances for water remediation. First, 4-NP was employed as a probe to characterize the catalytic activity of different Ag@Cu_2_O–Pt heterostructures. This classical reaction is based on the degradation of 4-NP by a nanocatalyst in the presence of NaBH_4_. This catalytic system is often employed to evaluate the catalytic performance of a nanostructure. UV-Vis spectroscopy was employed to monitor the production of 4-aminophenol which was reduced by 4-NP in the presence of catalyst (Ag@Cu_2_O and Ag@Cu_2_O–Pt). As shown in Appendix A, the catalytic reduction time increased significantly with increasing amounts of the Pt precursor. The catalytic time for Ag@Cu_2_O was 45 min (shown in Figure 5A), and those for Ag@Cu_2_O–Pt ranged from 28 to 20 min (shown in Appendix A). Compared with that of Ag@Cu_2_O, the catalytic efficiency of the Ag@Cu_2_O–Pt nanoparticles is higher. During the catalytic process, electrons are supplied by BH_4_^−^ to the catalyst such that 4-NP is adsorbed onto the catalyst. The abundance of Ag@Cu_2_O–Pt interfaces results in the high catalytic efficiency. An electron-depleted region near the surface migrates from Ag@Cu_2_O to Pt, forming an electron-rich region. These excess electrons exist on the Pt, which facilitates the absorption of electrons by the 4-NP molecules. If there are more interfaces, then there will be more areas with excess electrons. Therefore, when the amount of Pt nanoparticles coated on Ag@Cu_2_O increases, the catalytic activity also increases. When the amount of Pt precursor was increased to 1.58 × 10^−4^ and 1.63 × 10^−4^ M (Figure 5B), the reaction time was 20 min. It is most likely that, with an increase in precursor concentration, the density of the Pt nanoparticle coating increased. Therefore, catalytic activity of Pt-coated Ag@Cu_2_O increased significantly compared to that of the uncoated catalyst. The linear relationship between the absorption logarithm (ln (A)) and time *(t)*, shown in Figure 5B, indicates that the reduction of 4-NP by Ag@Cu_2_O–Pt conforms to pseudo-first-order kinetics.

### 3.3. Catalytic Degradation of Organic Dyes (MO)

To understand the catalytic effect of Ag@Cu_2_O–Pt on dye pollutants, MO was selected as the catalytic target. In this catalytic experiment, the catalytic effect of Ag@Cu_2_O–Pt is characterized by changes in the MO concentration. Under the action of NaBH_4_, the 460 nm absorption peak of MO gradually disappeared and a new absorption peak appeared at 260 nm, which indicates that MO had degraded and a new product had formed (Figure 6B). The Ag@Cu_2_O–Pt nanoparticles exhibit excellent catalytic performance compared with that of the Ag@Cu_2_O nanoparticles, which were barely able to catalyze MO (Figure 6A). This is because the porous structure of the Ag@Cu_2_O–Pt surface has good adsorptivity towards the dye molecules and better MO adsorption onto the catalyst surface facilitates completion of the catalytic process. Pt itself has a good catalytic performance, which promotes the reactivity of the catalytic process. Therefore, Pt-decorated Ag@Cu_2_O nanoparticles improved the catalytic activity compared to that of the uncoated nanoparticles.

## 4. Conclusions

We used chemical methods to synthesize Ag@Cu_2_O–Pt nanoparticles. Because of the potential difference between PtCl_6_^−^/Pt and Cu^2+^/Cu_2_O, PtCl_6_^−^ was employed as a precursor to grow Pt nanoparticles on the Ag@Cu_2_O surface to reduce the surface energy. With the changing amount of Pt precursor, it was easy to control the growth density of the Pt nanoparticles on the Ag@Cu_2_O surface. More importantly, the catalytic activity was strongly dependent on the Pt structure on the Ag@Cu_2_O. Besides that, 4-NP and MO were used as probes to evaluate the catalytic performance of Ag@Cu_2_O–Pt nanoparticles. The catalytic performance of Ag@Cu_2_O–Pt with different densities of Pt nanoparticles on the surface toward 4-NP and MO showed that the results require an understanding of the catalytic mechanism of the nanocomposite. The Ag@Cu_2_O–Pt nanocomposites exhibited potential for application in the catalytic treatment of water pollution.

## Figures and Tables

**Figure 1 molecules-24-02721-f001:**
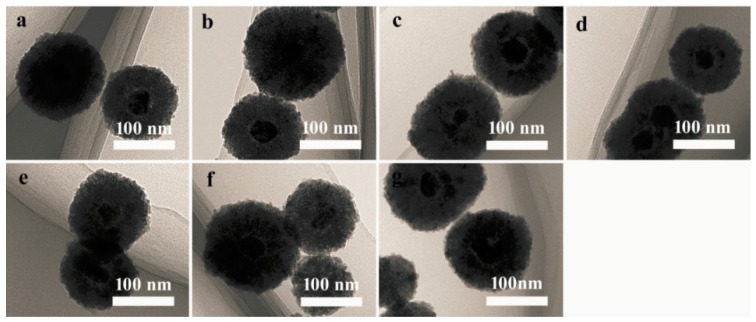
TEM images of Ag@Cu_2_O (**a**) and Ag@Cu_2_O–Pt with Pt precursor concentrations of 0.95 × 10^−4^ (**b**), 1.26 × 10^−4^ (**c**), 1.43 × 10^−4^ (**d**), 1.52 × 10^−4^ (**e**), 1.58 × 10^−4^ (**f**), and 1.63 × 10^−4^ (**g**) mol/L.

**Figure 2 molecules-24-02721-f002:**
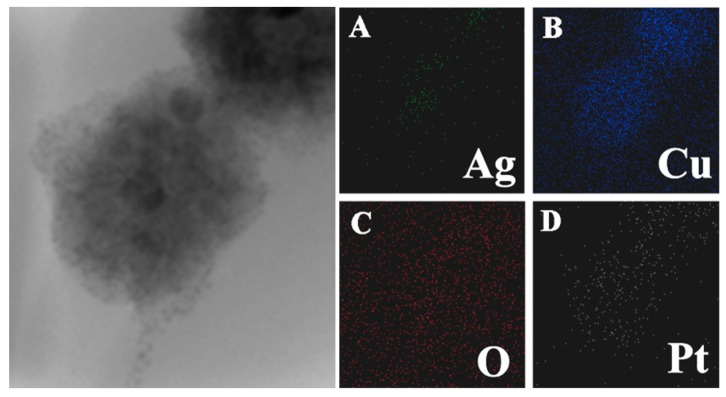
**A** TEM image of Ag@Cu_2_O–Pt with a Pt precursor concentration of 1.58 × 10^−4^ mol/L.

**Figure 3 molecules-24-02721-f003:**
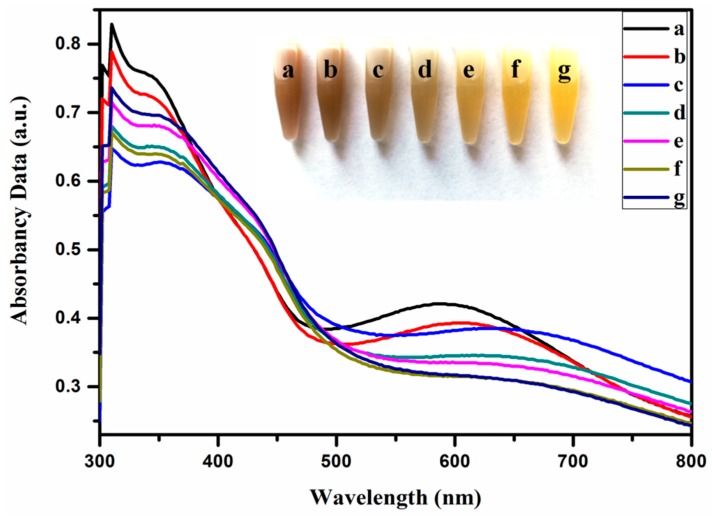
Visual color change upon the addition of increasing concentrations of HPtCl_6_ to Ag@Cu_2_O and the corresponding absorbance spectra: 0 (a), 0.95 × 10^−4^ (b), 1.26 × 10^−4^ (c), 1.43 × 10^−4^ (d), 1.52 × 10^−4^ (e), 1.58 × 10^−4^ (f), and 1.63 × 10^×4^ (g) mol/L.

**Figure 4 molecules-24-02721-f004:**
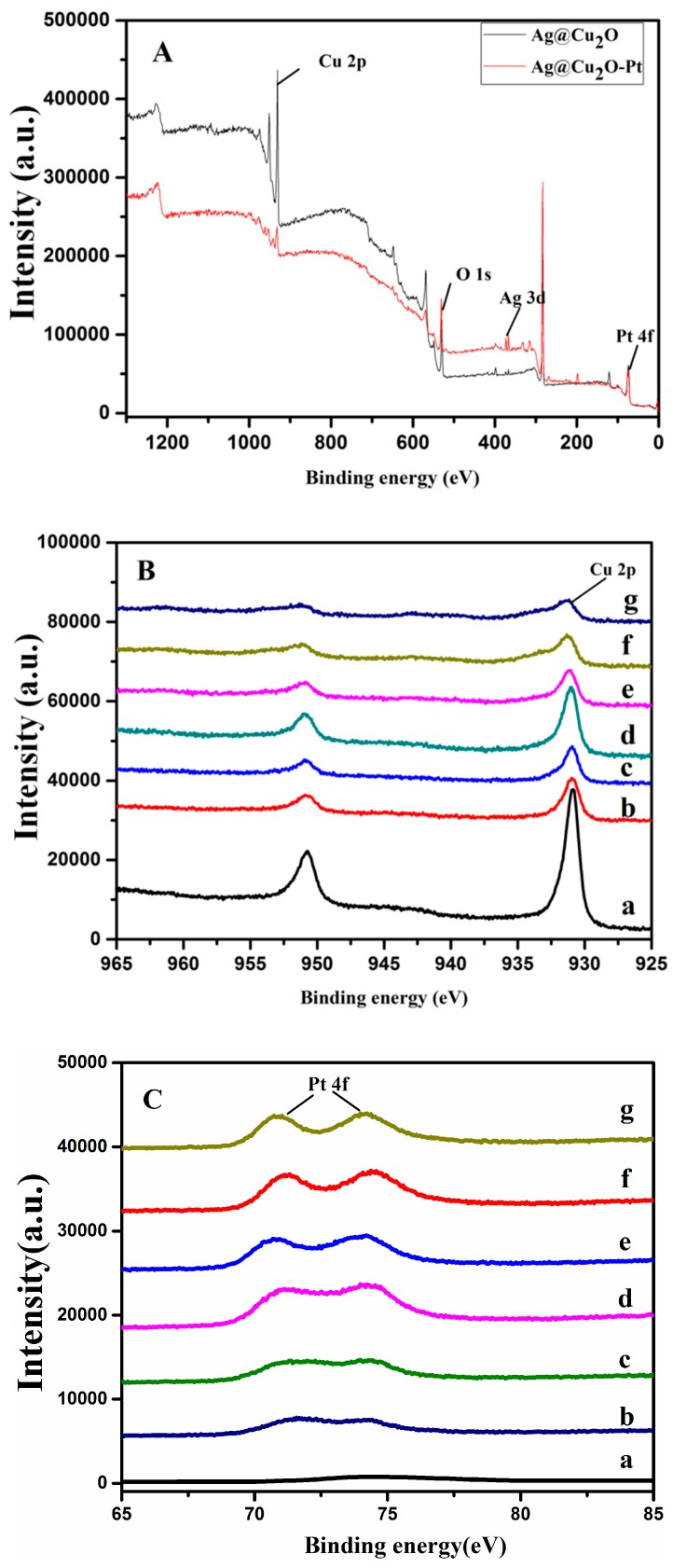
XPS spectra: (**A**) Survey spectra of Ag@Cu_2_O and Ag@Cu_2_O–Pt (prepared using a Pt precursor concentration of 1.58 × 10^−4^ M). (**B**) Cu 2p peak regions of Ag@Cu_2_O (a) and Ag@Cu_2_O–Pt (b–g) prepared with various concentrations of the Pt precursor (b: 0.95 × 10^−4^, c: 1.26 × 10^−4^, d: 1.43 × 10^−4^, e: 1.52 × 10^−4^, f: 1.58 × 10^−4^, and g: 1.63 × 10^−4^ mol/L). (**C**) Pt 4f peak regions of Ag@Cu_2_O (a) and Ag@Cu_2_O–Pt (b–g) (b: 0.95 × 10^−4^, c: 1.26 × 10^−4^, d: 1.43 × 10^−4^, e: 1.52 × 10^−4^, f: 1.58 × 10^−4^, and g: 1.63 × 10^−4^ mol/L).

**Figure 5 molecules-24-02721-f005:**
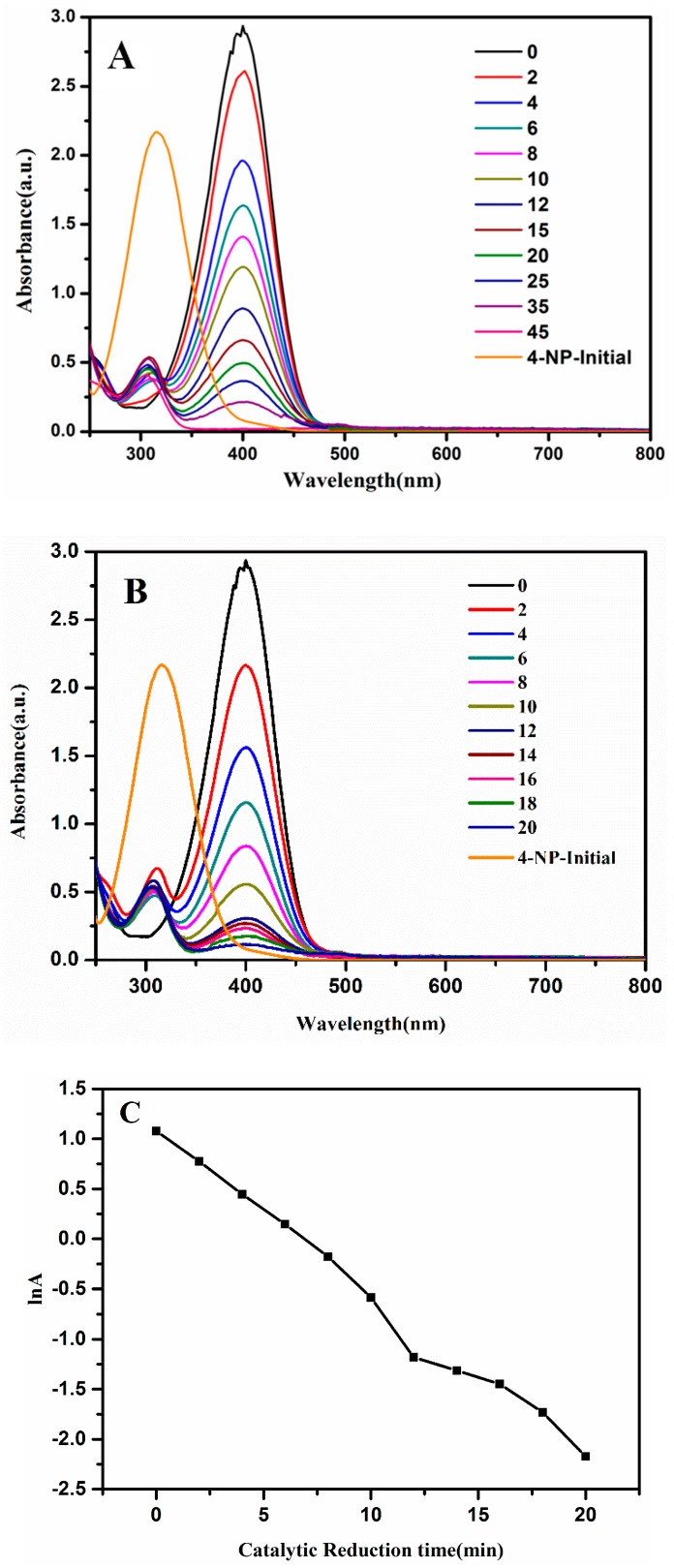
UV-Vis absorption spectra of the reduction of 4-nitrophenol (4-NP) by NaBH_4_ in the presence of (**A**) Ag@Cu_2_O (*t* = 45 min) and (**B**) Ag@Cu_2_O-Pt (*t* = 20 min). (**C**) The logarithm of the absorbance at 400 nm vs. the catalytic reduction time.

**Figure 6 molecules-24-02721-f006:**
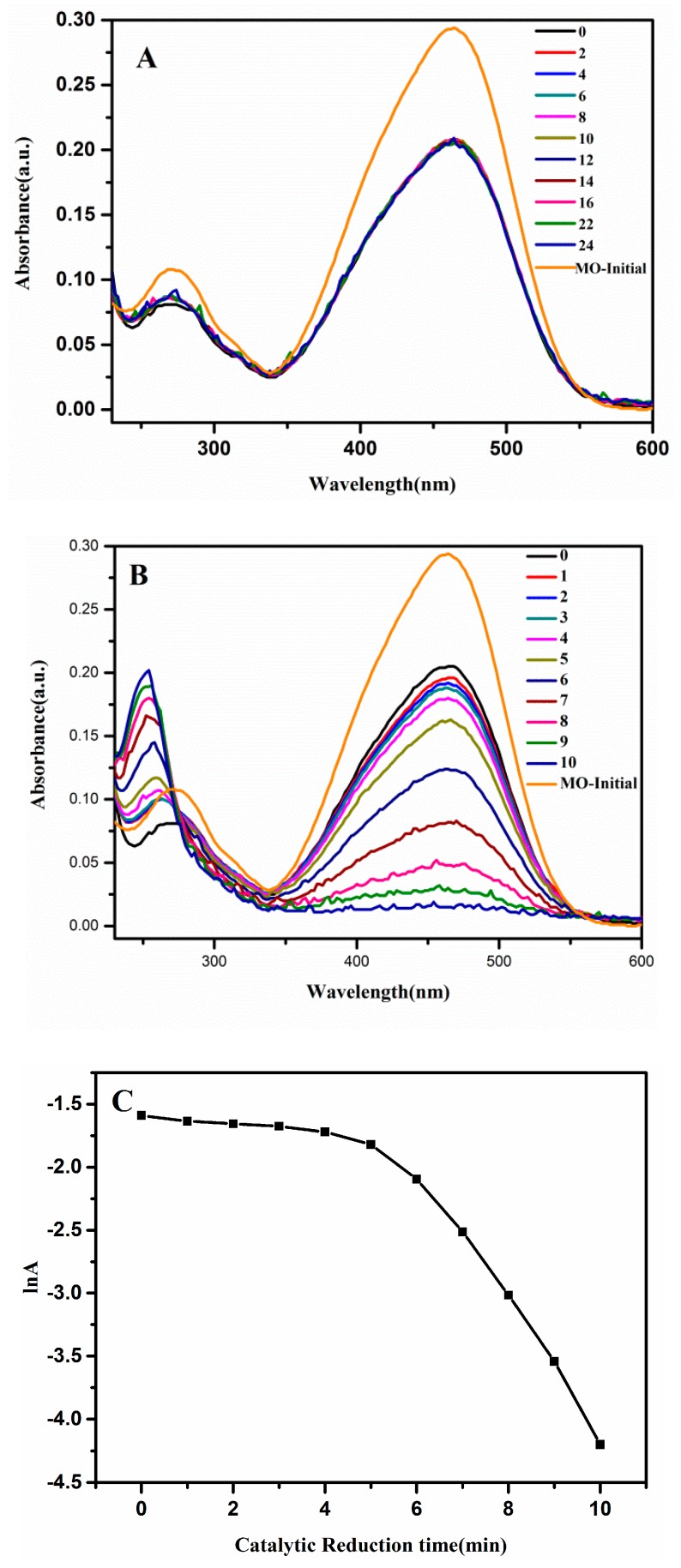
UV-Vis absorption spectra for the reduction of methyl orange (MO) by NaBH_4_ in the presence of (**A**) Ag@Cu_2_O (t = 24 min) and (**B**) Ag@Cu_2_O-Pt (t = 10 min). (**C**) The logarithm of the absorbance at 400 nm vs. the catalytic reduction time.

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
