# Peer review of "Catalysis of Organic Pollutants Abatement Based on Pt-Decorated Ag@Cu2O Heterostructures"

_molecules, 2019, doi:10.3390/molecules24152721_

Round 1

Reviewer 1 Report

The study addressed by the authors can be interesting but it in the present form the manuscript does not allow to appreciate it. In my opinion it must be rewritten with a thorough revision of the English expression and a revised introduction with a better structure and exposition line.  Regarding the scientific content, the authors must better define the reactions studied (in the introduction, the authors refer in general to catalysis) and also the composition and structure of the prepared samples. Better define the meaning of the symbols @ and – to identify the relative interaction of phases.  

p.2 how has been the size of the Ag particles measured?

p.3 line 93, the product is dissolved in ethanol??, line 105,  2 mL of particles? It is not clear if the concentration of the nanoparticles suspension is not given

p.4 Pt nanoparticles prepard by in situ substitution?

Figures 3 and 4 can be combined, besides they do not seem to be coherent.  In any case, how do the information obtained by this characterization technique support the obtained catalytic activity results?

The authors attribute, at least partially, the better behavior of the Pt containing samples to their higher porosity, however such a porosity is not explained nor proved by any characterization technique.

The UV-Vis data presented in Figures 6A (what do the orange and black lines represent?) and 7B to which Pt sample correspond?? It is not stated and besides there are not data shown corresponding to samples with different Pt content, that seems to be one of the main issues of the work, Some information on this appears in figure S1 but the caption is “Catalytic  reduction time vs. different….”  And it is not clear at all. This figure (well explained) is more important that showing the decrease in the intensity of the UV-Vis signals to give the kinetics obtained with ONE sample (but it is not indicated which one!)

Sentence in conclusions: “More importantly, the catalytic activity was strongly dependent on the Pt structure on the  Ag@Cu2O”, but no information on the Pt structure is given.

Author Response

Please see the attacment

Reviewer 2 Report

The paper presents interesting data related to the synthesis and catalysis by Pt-Ag-Cu2O nanoparticles. The authors succeeded in developing an original recipe for the synthesis of the Pt-decorated Ag-Cu2O composite nanoparticles. The method is not only simple and convenient, it is also very fast and allows the reduction of the preparation time. This is the strong site of the publication. The weakness of the paper is poor English and bad quality of figures. Thus, the following comments should be however taken into account in the paper revision:

1.      The title of the paper should be changed to "Catalysis of Organic Pollutants Abatement Based on Pt-Decorated Ag@Cu2O Heterostructures".

2.      The use of English language must be improved.

3.      The ordinate axis in Figures with XPS data should read Counts or Intensity.

4.      XPS spectra in the region of Pt binding energies should be presented (not just a survey spectrum).

5.      The curves corresponding to t=0 should be denoted in Figs. 6, 7 as 4-NP-Initial and MO-Initial

Round 2

Reviewer 2 Report

The authors revised the manuscript in agreement with the comments